# Sexual and Reproductive Outcomes in Obese Fertile Men with Functional Hypogonadism after Treatment with Liraglutide: Preliminary Results

**DOI:** 10.3390/jcm12020672

**Published:** 2023-01-14

**Authors:** Sandro La Vignera, Rosita A. Condorelli, Aldo E. Calogero, Rossella Cannarella, Antonio Aversa

**Affiliations:** 1Department of Clinical and Experimental Medicine, University of Catania, 95123 Catania, Italy; 2Department of Experimental and Clinical Medicine, University Magna Graecia of Catanzaro, 88100 Catanzaro, Italy

**Keywords:** male hypogonadism, metabolic hypogonadism, obesity, liraglutide, sexual function

## Abstract

**Purpose**: To prospectively investigate the effects of treatment with liraglutide, a glucagon-like peptide 1 (GLP1) analog, on reproductive and sexual function in men with metabolic hypogonadism who are of childbearing age. **Materials and Methods**: To accomplish this purpose, 110 men of childbearing age (18–35 years) with metabolic hypogonadism were enrolled and divided into three groups, according to their desire to have children. Group A was made up of men actively seeking fatherhood, Group B, of men who did not seek fatherhood, and Group C, of men who had already fathered a child. Group A patients were treated with gonadotropins (urofollitropin at 150 IU, three times a week, and human chorionic gonadotropin at 2000 IU, twice a week), Group B patients with liraglutide (3 mg daily), and Group C patients with transdermal testosterone (60 mg per day). All patients were treated for 4 months. **Results**: Patients treated with liraglutide (Group B) showed significant improvement in conventional sperm parameters, compared to baseline and Group A patients, and in the quality of erectile function compared to baseline and patients of Groups A and C. In addition, they had significantly higher levels of total testosterone and sex hormone-binding globulin serum levels after 4 months of treatment with liraglutide than those achieved by patients in the other two groups at the end of the respective treatments. Finally, Group B patients also showed significantly higher serum gonadotropin levels than the other groups. **Conclusions**: The results of this study showed, for the first time, the efficacy of liraglutide, a GLP1 analog, for the pharmacological treatment of male patients with metabolic hypogonadism. Liraglutide has also shown advantages over traditional treatments on both reproductive and sexual function and appears to offer greater benefits in terms of metabolic protection. These findings suggest that liraglutide is a useful drug for the treatment of obese males with metabolic hypogonadism.

## 1. Introduction

Obesity and metabolic diseases are able to alter testicular function in men of fertile age [1]. At the same time, hypogonadism worsens the evolution of obesity and metabolic diseases in young adults [2]. The link between testicular function and adipose tissue has been the subject of discussion for many years [3]. Although the triggering element is not clear, a bidirectional relationship has been widely demonstrated. Indeed, the reproductive and sexual dysfunction of obese patients is well documented. In particular, a large percentage of obese men have poor semen quality and erectile dysfunction [4]. From a pharmacological point of view, the efficacy of first-line drugs for the treatment of erectile dysfunction, such as selective phosphodiesterase 5 inhibitors (PDE5i), may be conditioned by the low levels of serum testosterone present in these patients [5]. Indeed, we know that in the absence of adequate circulating concentrations of androgens, the production of nitric oxide [6], as well as phosphodiesterase expression, are impaired [7]. Therefore, a large percentage of patients with hypogonadism have a poor response to these types of molecules.

At the same time, the hormonal treatment of patients with oligozoospermia and normal or low serum gonadotropin levels, which are frequently present in male obese patients, could theoretically take advantage of a combination of follicle-stimulating hormone (FSH) and human chorionic gonadotropin (hCG). However, the efficacy of this treatment and the dosage of chorionic gonadotropin to be used are controversial [8]. Furthermore, the use of testosterone-based preparations in men seeking fatherhood is considered detrimental to semen quality because it suppresses gonadotropin release [9]. The ideal treatment for these forms of hypogonadism, with predominantly central etiopathogenesis (low serum total testosterone (TT) and normal/low serum gonadotropin levels), should be able to increase testosterone levels without suppressing the hypothalamic-pituitary-testicular axis, while maintaining a safety profile on other parameters, such as prostate-specific antigen (PSA) levels and hematocrit, alterations in which represent the main problem with an excess of hormone replacement therapy.

Thus, this study aimed to evaluate the effects of treatment with the glucagon-like peptide 1 (GLP1) analog liraglutide, at a dosage of 3.0 mg, on sperm parameters, the quality of their erectile function, and on some parameters of clinical safety (PSA and hematocrit levels) in obese young men with metabolic hypogonadism who are of childbearing age.

## 2. Materials and Methods

A total of 110 men aged 18 to 35 years (mean age: 26 ± 6) were consecutively enrolled, after signing informed consent, according to the following inclusion criteria:Moderate and/or severe obesity (body mass index (BMI) of between 30 and 39.99);Waist circumference >94 cm;Severe erectile dysfunction ((ED)—International Index of Erectile Function 5 Items (IIEF-5) score below 7) that is unresponsive to conventional treatment with PDE5i during the last 3 months, according to a previously published procedure [10];TT < 12 nmol/L according to the Italian Society of Endocrinology guidelines [11];Normal–low gonadotropin levels (lowest quartiles of the reference range);Levels of sex hormone-binding globulin (SHBG) that are reduced compared to the reference range (confirmed data);Homeostasis Model Assessment Index (HOMA) > 2.5 [12].

All enrolled patients were divided into three treatment groups, according to their intent to achieve fatherhood, and underwent a standard low-carbohydrate diet (1400–1800 Kcal/day):Group A (*n* = 35): Patients with an active desire for fatherhood. They were prescribed conventional treatment for fertility using urofollitropin 150 IU subcutaneously (three times a week) associated with hCG 2000 IU subcutaneously (twice a week) for 4 months;Group B (*n* = 35): Patients with no active desire for fatherhood. They were prescribed liraglutide 0.6 mg subcutaneously every day for the first week, then 1.2 mg (second week), 1.8 mg (third week), 2.4 mg (fourth week), and 3.0 mg (from the fifth week for another three months);Group C (*n* = 40): Patients who had already fathered a child and were not seeking fertility. They were prescribed transdermal testosterone gel (2%) at 60 mg every day for 4 months.

Exclusion criteria:Patients with primary testicular disease: cryptorchidism, testicular tumor, varicocele (all grades) or previous varicocelectomy, previous orchitis, or previous testicular torsion;Patients with increased levels of prolactin or TSH;Patients with hypothalamic-pituitary lesions described via magnetic resonance imaging;Patients with arterial hypertension, dyslipidemia, impaired fasting blood glucose, or diabetes mellitus.

Parameters that were prospectively evaluated before and after drug treatment:Weight, height, BMI, and waist circumference;HOMA index, TT, LH, FSH, SHBG, total PSA, and hematocrit;Semen analysis (except for group C patients);IIEF-5 and the frequency of PDE5i use, as needed, for sexual activity.

### 2.1. Hormone Measurements

Hormones were measured in the blood that was withdrawn between 8:00 and 9:00 a.m., in fasting conditions. Electrochemiluminescence (Hitachi-Roche equipment, Cobas 6000, Roche Diagnostics, Indianapolis, IN, USA) was used to measure gonadotropin, TT, and SHBG levels. We used the following laboratory reference values to distinguish normal from abnormal values: LH 1.14–8.75 IU/L, FSH 0.95–11.95 IU/L, TT 0.478–9.8 ng/mL, and SHBG 18.3–54.1 nmol/L.

### 2.2. Semen Analysis

Sterile containers were used to collect the semen specimens, following 2–7 days of sexual abstinence. The analysis was performed soon after liquefaction and all the parameters were evaluated according to the 2021 WHO guidelines, as detailed elsewhere [13]. 

### 2.3. Statistical Analysis

Continuous data are reported as mean ± standard deviation (SD). The variables’ distribution was assessed with the Shapiro–Wilk test. The one-way analysis of variance (ANOVA), followed by the Tukey–Kramer post hoc test, was used to analyze the within-group differences. The chi-squared test was adopted to assess any difference in the frequency of PDE5i use among the groups. MedCalc software (MedCalc Ltd., Ostend, Belgium), version 19.6—64 bit, was used to perform the analysis. A *p*-value of < 0.05 was considered statistically significant.

### 2.4. Ethical Approval

The study was conducted according to the guidelines of the Declaration of Helsinki and was approved by the internal Institutional Review Board of the Division of Endocrinology, Metabolic Diseases, and Nutrition, at the teaching hospital of “G. Rodolico–San Marco” of the University of Catania (Catania, Italy) (protocol code 01/2022 approved on 30 January 2022). 

## 3. Results

The baseline anthropometric and biochemical parameters of the three groups of patients are shown in Table 1. No significant differences were found between groups at baseline. After treatment, the patients of group B showed the best outcomes for all the studied parameters. In particular, they showed a significant reduction in body weight of 10.3% (116 ± 10 vs. 104 ± 6 Kg, *p* < 0.05), a BMI reduced by 16.7% (36 ± 3 vs. 30 ± 2 Kg/m^2^, *p* < 0.05), and a waist circumference reduction of 8.3% (108 ± 6 vs. 99 ± 4 cm, *p* < 0.05), with a statistically significant difference from baseline and from what was achieved by the patients of the other two groups at the end of their specific treatments (Figure 1).

After treatment, group B showed serum TT levels that were significantly increased by 192.9% (1.4 ± 0.6 vs. 4.1 ± 0.5 ng/mL, *p* < 0.05), and SHBG levels increased by 157.1% (14.0 ± 3.0 vs. 36.0 ± 4.0 nmol/L, *p* < 0.05) compared to baseline and compared to the levels found in the other two groups at the end of treatment. Interestingly, FSH (Group A: 0.9 ± 0.2 IU/L vs. Group B: 2.6 ± 0.2 IU/L vs. Group C: 0.2 ± 0.1 IU/L, *p* < 0.05) and LH (Group A: 1.0 ± 0.3 IU/L vs. Group B: 3.2 ± 0.2 IU/L vs. Group C: 0.3 ± 0.1 IU/L, *p* < 0.05) plasma levels significantly increased in Group B compared with other groups, along with a reduction in the HOMA index (Group A: 5.2 ± 0.6 vs. Group B: 3.1 ± 0.6 vs. Group C: 4.8 ± 1.2, *p* < 0.05) (Figure 2A–C).

After treatment, Group B patients showed a significant increase in all conventional sperm parameters. In particular, the increase in sperm motility after treatment with liraglutide was highly significant (14 ± 2 vs. 34 ± 4%, *p* < 0.05) and was equal to 142.9%. As expected, Group A also showed a 58.3% increase in sperm motility compared to baseline values (12 ± 3 vs. 19 ± 2%, *p* < 0.05) (Figure 3).

Surprisingly, group B patients showed a significantly higher IIEF-5 score (4 ± 2 vs. 21 ± 4, *p* < 0.05) than the other two groups at the end of treatment. The frequency of using PDE5i for erectile dysfunction was significantly reduced in group B patients during 4 months of treatment with liraglutide (Group A: 48%, Group B: 31%, and Group C: 45%) (Figure 4).

## 4. Discussion

The results of the present study suggest, for the first time, that patients with metabolic hypogonadism, characterized by low serum TT and SHBG levels, low–normal gonadotropins, BMI > 30 kg/m^2^, waist circumference > 94 cm, HOMA index > 2.5, showed improvement in sperm parameters and sexual function and returned to being eugonadal after four months of liraglutide treatment. In particular, after therapy, the improvement in conventional sperm parameters and erectile function, assessed by the validated IIEF-5 questionnaire, was greater in those patients treated with this GLP1 analog than in patients treated with gonadotropins or transdermal testosterone.

The role of GLP-1 in energy balance has already been recognized. It is exerted via binding to a well-characterized GLP-1 receptor (GLP1R) [14]. Current data indicate the presence of several GLP1R-positive neuronal populations that contribute to the anorectic effects of GLP1R agonists (GLP1-Ra). Indeed, an infusion of GLP1 causes a reduced perception of hunger and an increased sensation of fullness in healthy non-obese volunteers. These effects have been associated with decreased activation of the amygdala, caudate, insula, nucleus accumbens, orbitofrontal cortex, and putamen [15], although further effects mediated by leptin and its receptor have also been recognized [16,17,18].

Hyperglycemia and insulin resistance in patients with obesity and type 2 diabetes mellitus (T2DM) affect the function of the hypothalamic-pituitary-testicular axis in men, leading to a particular form of hypogonadotropic hypogonadism, known as metabolic hypogonadism, characterized by low testosterone levels and inappropriately normal gonadotropins [19]. Lifestyle changes are known to be very effective in improving testosterone levels in men with metabolic hypogonadism [20]. However, the effects of GLP1-Ra on male metabolic hypogonadism are still poorly studied. Although an observational and retrospective study by Giagulli and colleagues on hypogonadal patients with T2DM showed that testosterone replacement therapy (TRT), when given together with weight-reducing anti-hyperglycemic compounds (in particular, GLP1-Ra), was effective not only in maintaining stable serum TT levels in the normal range for young healthy men but also in reducing hyperglycemia and hyperlipidemia to achieve their treatment targets, and in reducing weight to considerably ameliorate the severity of erectile dysfunction [21]. In another study, Jensterle and colleagues demonstrated that a 16-week treatment with liraglutide (3.0 mg/day) in obese men who responded poorly to lifestyle changes led to robust weight loss, compared with patients treated with transdermal testosterone. Furthermore, liraglutide treatment significantly increased the serum TT, LH, and FSH levels; this was associated with improved sexual function [22]. According to these findings, the present study showed that treatment with liraglutide rapidly restored their eugonadal state, along with an increase in serum TT levels, recovery of the hypothalamic-pituitary-testicular axis function, reduction of the anthropometric measures of obesity and of the HOMA index, and a significant improvement in sperm parameters and sexual function. 

According to previous evidence, the increase in testosterone levels could be due to the weight-reducing effect of GLP1-Ra by lowering food intake, more than the effects on gonadotrophin secretion [21,23]. Consequently, the study by Giagulli and colleagues reported no increase in gonadotropin levels after a 12-month administration of liraglutide in 16 diabetic patients with functional hypogonadism. However, as stated by the same authors, the lack of serum gonadotropin increase is surprising, since the increase in testosterone levels can likely be due to the effects of weight reduction on GnRH-secreting neurons. Therefore, they ascribed the failure to find an LH and FSH increase to using a single gonadotropin measurement, which makes it impossible to identify their pulse peak [21]. In contrast, another study reported no increase in gonadotropin pulse rate after an 8-hour intravenous infusion of 0.8 pmol/kg/min of GLP1 [23]. The latter study was unable to demonstrate any direct effect of GLP1 on the GnRH-secreting neurons, but this study reports no evidence of a possible indirect effect, such as, for example, being mediated by weight loss [23]. Therefore, based also on our findings, a weight-loss-mediated effect of GLP1-Ra on GnRH-secreting neurons should not be excluded. 

A possible direct effect of liraglutide in the modulation of testicular function cannot be excluded, this being supported by the recent identification of GLP1R in healthy (non-tumor) human Leydig cells. In particular, the positive effects of GLP1-Ra on erectile function have also been attributed to the direct effect of these drugs on the endothelium [24]. Testosterone levels are, in some way, one of the links between internal medicine and sexual medicine. These levels influence endothelial function, as well as susceptibility to the development of comorbidities with aging [25]. Testosterone is involved in regulating the balance between endogenous vasoconstrictors and the vasorelaxant agents of vascular tone [26]. Consequently, androgen receptors are expressed in the vascular endothelium and in smooth muscle cells. Thus, testosterone can directly influence vascular tone via several pathways [27]. Indeed, it has been shown to regulate coronary, aortic, and brachial vasculature dilatation [28]. Therefore, the increase in endogenous testosterone levels observed in the present study (compared to TRT) may have produced penile vascular improvements in the short term regarding sexual function, bearing in mind that the genomic effects of exogenous TRT can take from six to twelve months to occur [29].

Notably, normal serum testosterone levels were not achieved in patients in Group A, who were treated with gonadotropins. hCG is usually administered starting at 1000 IU twice a week. The dose can be increased to 2000, 3000, 4000, or 5000 IU two or three times a week until normal testosterone levels are achieved [30]. The dose chosen in our study was 2000 IU twice a week, which represents an intermediate dosage used in patients who desire fatherhood. This may explain why normal testosterone levels were not achieved. In particular, hCG has a relatively short half-life, justifying the need to administer it two to three times a week. In the present study, testosterone levels were measured just prior to drug administration, i.e., when blood levels were at their lowest. This could further explain why normal testosterone levels were not achieved.

It is increasingly recognized that SHBG, which is produced by the liver, is involved in the development of metabolic disorders [31]. Only a single study carried out in women has evaluated the relationship between SHBG and non-alcoholic fatty liver disease (NAFLD), showing an association between low SHBG levels and the risk of NAFLD [32]. Interestingly, this study reported an association between SHBG levels of <30 nmol/L and a fivefold higher risk of NAFLD, compared with patients with SHBG levels of >60 nmol/L [32]. Another study has corroborated this interesting finding in the female population, identifying an SHBG threshold of 33.4 nmol/L as the most appropriate level by which to identify women with pathologic NAFLD [33]. At present, only one study has been conducted in men with low serum testosterone levels. The authors found higher noninvasive indices of NAFLD, but not liver fibrosis, compared to men with higher testosterone levels. Therefore, they concluded that fatty liver indices were inversely and independently associated with testosterone levels, without investigating the SHBG levels [34]. In our study, liraglutide produced a marked increase in SHBG levels, taking them well above the identified cut-off for NAFLD, thus suggesting a protective effect on the liver that is not a predominant short-term feature of either hCG administration or TRT.

A possible explanation for the effects on the Sertoli cell and sperm parameters is that severe obesity may trigger the activation of the NLRP3 inflammasome complex, leading to the production of the proinflammatory cytokines, IL-1β and IL-18 [35]. In the seminal plasma, the main radical oxygen species (ROS) that are present are superoxide, hydroxyl, and hydrogen hydroxide radicals, the content of which is regulated in the male reproductive tract by defensive mechanisms that control the balance between the formation and elimination of reactive oxygen and nitrogen species, to preserve the normal activity of somatic and germ cells [36]. Interestingly, a study on high-fat diets (administered in an HFD-induced obese mouse model proposed by Fan and colleagues) demonstrated that a prolonged high-fat diet could lead to increased expression levels of the NLRP3 inflammasome and proinflammatory cytokines, such as IL-6 and TNF-α, in the testis, epididymal caput, epididymal cauda, prostate, and seminal vesicles, influencing testicular function. Furthermore, their clinical experiments conducted on overweight/obese patients revealed poor semen quality, combined with higher levels of IL-6 and TNF-α in the seminal plasma. These data suggest that NLRP3 inflammasome activation strongly contributes to the chronic sterile testicular inflammation observed in obese men, leading to infertility [37]. Under normal conditions, physiological levels of ROS are important for several male reproductive functions, such as spermatogenesis, sperm viability and motility, and acrosome reaction. However, when the intratesticular ROS content increases, thioredoxin-interacting protein activation induces an inflammatory state with a lethal outcome for cell survival, leading to the release of IL-1β and IL-18 and caspase-1 cleavage [38]. While the present study did not investigate conventional inflammatory parameters (neither primary nor secondary outcomes), we can argue that many of the effects found may have been mediated by reduced inflammation following significant weight loss.

Our study has some limitations. Our conclusions are limited by the small sample size, evaluated without a placebo, and also by the absence of controlled physical exercise as an add-on treatment for obesity. Furthermore, no vascular study of the penile arteries was performed in this cohort of patients, mainly claiming disabilities related to excess body weight and fertility issues. Further, larger, and longer-duration studies should be powered using these preliminary results. We hypothesized that a complex bidirectional relationship between insulin resistance and metabolic hypogonadism was controlled in one direction. However, the main strength of this preliminary study is the original idea of comparing hormone replacement therapy and liraglutide in terms of changing body weight and recovering eugonadism, without the inhibition of gonadotropin levels and with a significant increase in the SHBG levels obtained after treatment with this GLP1-Ra.

## 5. Conclusions

In conclusion, this is the first clinical study comparing different treatments for metabolic hypogonadism in obese men of childbearing age. The results of the study showed that the GLP1-Ra liraglutide was found to be very effective, thus suggesting that GLP1-Ra can be used for bodyweight reduction and recovery of the testicular function in terms of reproductive and sexual function [39]. Additional placebo-controlled studies are ongoing to verify the preliminary results obtained in this study.

## Figures and Tables

**Figure 1 jcm-12-00672-f001:**
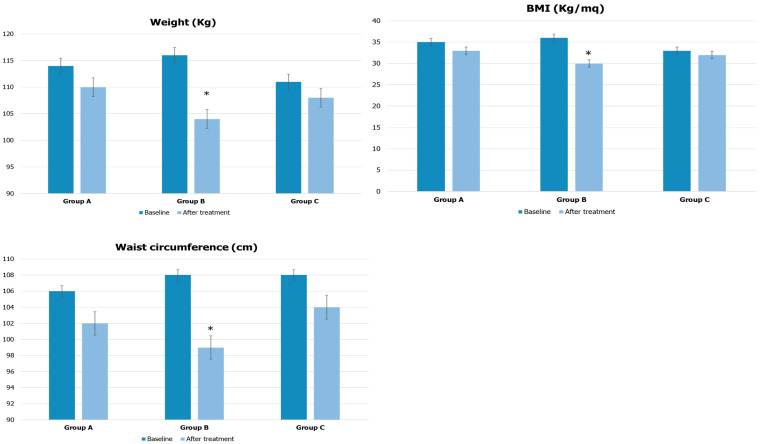
Anthropometric parameters before and after treatment in the three groups of patients. * *p* < 0.05 compared to baseline and to the other groups (baseline and after treatment).

**Figure 2 jcm-12-00672-f002:**
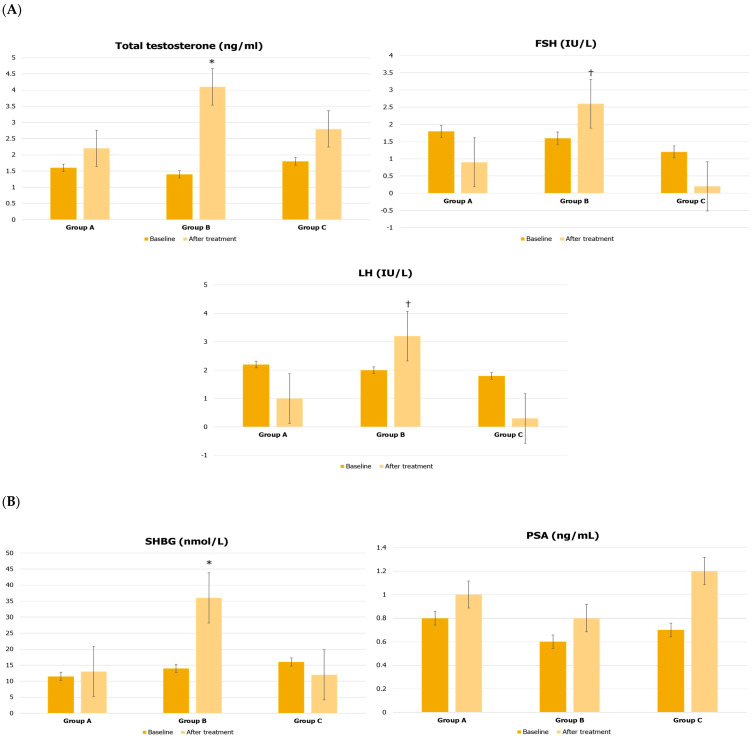
Laboratory evaluations before and after treatment in patients treated with urofollitropin plus hCG (Group A), liraglutide (Group B), or transdermal testosterone (Group C). * *p* < 0.05 compared to baseline and to the other groups (baseline and after treatment); † *p* < 0.05 compared to values of other groups after treatment.

**Figure 3 jcm-12-00672-f003:**
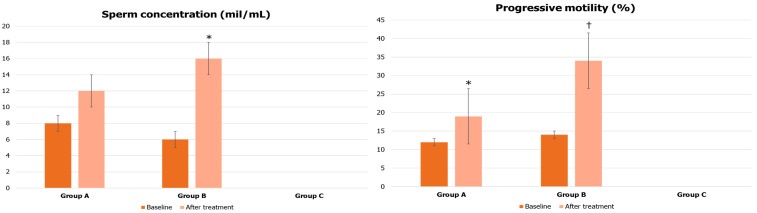
Conventional sperm parameters before and after treatment in patients treated with urofollitropin plus hCG (Group A), or liraglutide (Group B). Data on Group C (treated with transdermal testosterone) were not collected. * *p* < 0.05 compared to baseline (same group); † *p* < 0.05 compared to values of other groups after treatment.

**Figure 4 jcm-12-00672-f004:**
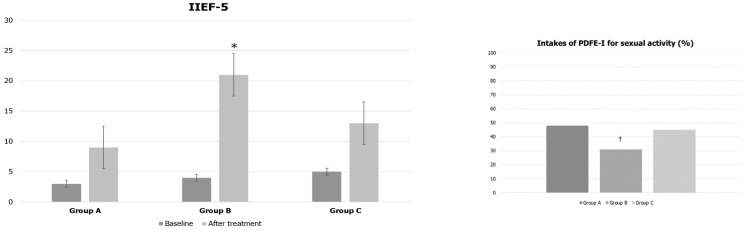
IIEF-5 and frequency of PDE5i use before and after treatment in patients treated with urofollitropin plus hCG (Group A), liraglutide (Group B), or transdermal testosterone (Group C). * *p* < 0.05 compared to baseline and the other values of the groups (at baseline and after treatment); † *p* < 0.05 compared to the other values of the other groups.

**Table 1 jcm-12-00672-t001:** Anthropometric and biochemical parameters in the three groups of patients at baseline.

Parameters	Group A	Group B	Group C
Weight (kg)	114 ± 6	116 ± 10	111 ± 5
BMI (kg/m^2^)	35 ± 3	36 ± 3	33 ± 2
Waist circumference (cm)	106 ± 5	108 ± 6	108 ± 4
Total testosterone (ng/mL)	1.6 ± 0.2	1.4 ± 0.6	1.8 ± 0.4
Follicle-stimulating hormone (IU/L)	1.8 ± 0.2	1.6 ± 0.3	1.2 ± 0.4
Luteinizing hormone (IU/L)	2.2 ± 0.2	2.0 ± 0.3	1.8 ± 0.4
Sex hormone-binding globulin (nmol/L)	11.5 ± 3.0	14.0 ± 3.0	16 ± 6.0
Prostate-specific antigen (ng/mL)	0.8 ± 0.2	0.6 ± 0.3	0.7 ± 0.1
Hematocrit (%)	40 ± 4	38 ± 6	42 ± 4
HOMA index	6.0 ± 1.2	6.5 ± 1.9	5.6 ± 1.6

## Data Availability

Data are available upon request to the corresponding author.

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
