# Peer review of "Sexual and Reproductive Outcomes in Obese Fertile Men with Functional Hypogonadism after Treatment with Liraglutide: Preliminary Results"

_jcm, 2023, doi:10.3390/jcm12020672_

Round 1
Reviewer 1 Report
This study evaluates the effect of liraglutide at the dosage of 3 mg on reproductive and sexual function in men with metabolic hypogonadism adding new interesting results in this field and confirming the potential efficacy of this type of treatment in this setting. Moreover, these data add new results showing a better outcome in terms of seminal parameters compared to the conventional treatment for fertility. Therefore all these data strengtheners the potential use of this drug in male reproductive functional disorders.
Few issues need to be addressed.
Minor comments
-
The authors should add a table with the anthropometric and biochemical baseline parameters of all the three groups.
-
It would be better for the reader to clarify the results adding the numerical data obtained for each group (Please provide in the text media, SD and/or range, as well as the P values). I suggest also to add the average lost or increase of the main anthropometric, hormonal and seminal parameters.
-
In group A, on gonadotropin treatment, normal T levels were not achieved. Can the authors add any comment of these results in the text/discussion?
-
Have the authors analysed the free-T levels? Can the authors provide those data in the text?
-
Group C on T treatment should not be considered for the sperm parameters because of absence of data (as correctly mentioned in the material and methods). Please eliminate group c in the figure and in the legend or add an explanation in the legend. (as reported is confusing)
-
Figure 3. Please provide another symbol for the P value < 0.01. The symbol used is the same one previously used in different way (for comparison between groups).
Author Response
Answers to the Reviewer #1’s comments
Manuscript ID jcm-2131925 - Revised
Comment 1: The authors should add a table with the anthropometric and biochemical baseline parameters of all the three groups.
Answer to comment 1: Added as requested (please see lines 132-139 and Table 1).
Comment 2: It would be better for the reader to clarify the results adding the numerical data obtained for each group (Please provide in the text media, SD and/or range, as well as the P values). I suggest also to add the average lost or increase of the main anthropometric, hormonal and seminal parameters.
Answer to comment 2: This information was added in lines 135-137, 147-154, 171-174, and 183-186.
Comment 3: In group A, on gonadotropin treatment, normal T levels were not achieved. Can the authors add any comment of these results in the text/discussion?
Answer to comment 3: We appreciated this comment. We modified the Discussion adding the following paragraph in lines 264-273: “Notably, normal serum testosterone levels were not achieved in patients of the Group A, that were treated with gonadotropins. hCG is usually given starting at 1,000 IU twice a week. The dose can be increased to 2,000, 3,000, 4,000, or 5,000 IU two or three times a week, until normal testosterone levels are achieved [30]. The dose chosen in our study was 2,000 IU twice a week, which represents an intermediate dosage used in patients who desire fatherhood. This may explain why normal testosterone levels were not achieved. In particular, hCG has a relatively short half-life, justifying the need to administer it two to three times a week. In the present study, testosterone levels are measured just prior to drug administration, i.e. when blood levels are at their lowest. This could further explain why normal testosterone levels were not achieved”.
- Salonia A, Bettocchi C, Boeri L, Capogrosso P, Carvalho J, Cilesiz NC, Cocci A, Corona G, Dimitropoulos K, Gül M, Hatzichristodoulou G, Jones TH, Kadioglu A, Martínez Salamanca JI, Milenkovic U, Modgil V, Russo GI, Serefoglu EC, Tharakan T, Verze P, Minhas S; EAU Working Group on Male Sexual and Reproductive Health. European Association of Urology Guidelines on Sexual and Reproductive Health-2021 Update: Male Sexual Dysfunction. Eur Urol. 2021;80(3):333-357.
Comment 4: Have the authors analysed the free-T levels? Can the authors provide those data in the text?
Answer to comment 4: Unfortunately, we do not have these values as, in clinical practice, we do not request direct measurement of free-T levels according to the most relevant guidelines (see for example those of the Endocrine Society). Since we did not request albumin measurements, we cannot provide calculated free-T levels.
Comment 5: Group C on T treatment should not be considered for the sperm parameters because of absence of data (as correctly mentioned in the material and methods). Please eliminate group c in the figure and in the legend or add an explanation in the legend. (as reported is confusing)
Answer to Comment 5: Done as requested (please see lines 176-178).
Comment 6: Figure 3. Please provide another symbol for the P value <0.01. The symbol used is the same one previously used in different way (for comparison between groups).
Answer to comment 6: Done, thank you – please see the revised Figure 3.
Reviewer 2 Report
Dear Author,
While your work is interesting, it contains several points that require a more thorough discussion and a deep revision before JCM can publish it.
Major points.
1. Given that your participants did not suffer from any dysmetabolic conditions such as metabolic syndrome, dyslipidemia, or overt diabetes mellitus, I recommend changing your work title as follows: Sexual and reproductive outcomes in obese fertile men with secondary hypogonadism after treatment with liraglutide: preliminary results.
2. It is not clear if your study is a prospective or retrospective study. Could you be more precise in saying the nature of the survey?
3. In your text, there are no data about the characteristics of hormonal methods implied. Further, you reported the reference values without saying if they belong to those reported by kits or are those referred to fertile and eugonadic local males. Please, add the intra and inter-coefficient variations of your methods and the normal values of the methods that you used for this study.
4. Although the sperm value of group C is not known, they can be considered fertile since they had fathered children. While both groups A and B showed low concentrations of sperm, it would be interesting to evaluate any statistical differences between other clinical and hormonal parameters among groups A, B, and C in basal conditions.
Minor points
1. Abstract: page 1, line 2. In your text, you reported that group C did not perform the sperm analysis. Therefore, at the end of the sentence, you need to delete group C.
2. Materials and methods: page 2, line 7. Could you add the reference to HOMA?
3. Discussion: page 7; line 2. Could you change the sentence as follows? The increase in testosterone levels can be due to the weight-reducing effect of GLP1-Ra by lowering food intake more than the effects of both gonadotrophins secretion (Giagulli VA et al Andrology. 2020 May;8(3):654-662. doi: 10.1111/andr.12754; Izzi-Engbeaya et al JCEM, 2020,105(4), 1119-25). Based on that evidence, moreover, would you discuss how and why liraglutide rose both gonadotropins in your study?
Author Response
Answers to the Reviewer #2’s comments
Manuscript ID jcm-2131925
Comment 1: Given that your participants did not suffer from any dysmetabolic conditions such as metabolic syndrome, dyslipidemia, or overt diabetes mellitus, I recommend changing your work title as follows: Sexual and reproductive outcomes in obese fertile men with secondary hypogonadism after treatment with liraglutide: preliminary results.
Answer to comment 1: We would prefer to replace the term “secondary” of the title you suggested with “functional”.
Comment 2: It is not clear if your study is a prospective or retrospective study. Could you be more precise in saying the nature of the survey?
Answer to comment 2: This is a prospective study. We have clarified it in lines 10 and 100.
Comment 3: In your text, there are no data about the characteristics of hormonal methods implied. Further, you reported the reference values without saying if they belong to those reported by kits or are those referred to fertile and eugonadic local males. Please, add the intra and inter-coefficient variations of your methods and the normal values of the methods that you used for this study.
Answer to comment 3: Electrochemiluminescence was used for hormonal measurements (please see lines 105-106). The technical information is provided in the manufacturer instructions (Hitachi-Roche equipment, Cobas 6000, Roche Diagnostics, Indianapolis, IN, USA). The reference values are those reported by the laboratory and were the same that we used as the normal range. Unfortunately, we do not have intra- and inter-coefficient variations.
Comment 4: Although the sperm value of group C is not known, they can be considered fertile since they had fathered children. While both groups A and B showed low concentrations of sperm, it would be interesting to evaluate any statistical differences between other clinical and hormonal parameters among groups A, B, and C in basal conditions.
Answer to comment 4: We appreciated your suggestion. An assessment of any difference between baseline values was performed. We found no differences between groups at baseline. Regarding anthropometric and biochemical parameters, please see Table 1 of the revised manuscript. As far as sperm parameters and IIEF-5 score at baseline, please see the following:
|
Parameter |
Group A |
Group B |
Group C |
|
Sperm concentration at baseline (mil/ml) |
8±4 |
6±2 |
Not Evaluated |
|
Progressive motility at baseline (%) |
12±3 |
14±2 |
Not Evaluated |
|
Normal forms at baseline (%) |
3±1 |
4±2 |
Not Evaluated |
|
IIEF- 5 at baseline |
3±2 |
4±2 |
5±1 |
As you can see, no difference is reported.
Comment 5: Abstract: page 1, line 2. In your text, you reported that group C did not perform the sperm analysis. Therefore, at the end of the sentence, you need to delete group C.
Answer to comment 5: Since the IIEF-5 score was evaluated in Group C, the sentence was rephrased as follows: “Patients treated with liraglutide (Group B) showed significant improvement in conventional sperm parameters compared to baseline and Group A, and quality of erectile function compared to baseline and patients in Groups A and C” (see line 20).
Comment 6: Materials and methods: page 2, line 7. Could you add the reference to HOMA?
Answer to comment 6: Done as requested (please see line 76).
Comment 7: Discussion: page 7; line 2. Could you change the sentence as follows? The increase in testosterone levels can be due to the weight-reducing effect of GLP1-Ra by lowering food intake more than the effects of both gonadotrophins secretion (Giagulli VA et al Andrology. 2020 May;8(3):654-662. doi: 10.1111/andr.12754; Izzi-Engbeaya et al JCEM, 2020,105(4), 1119-25). Based on that evidence, moreover, would you discuss how and why liraglutide rose both gonadotropins in your study?
Answer to comment 7: We appreciated this suggestion. This concept has now been expanded in lines 216-248.